# Interplay between Single-Ion and Two-Ion Anisotropies in Frustrated 2D Semiconductors and Tuning of Magnetic Structures Topology

**DOI:** 10.3390/nano11081873

**Published:** 2021-07-21

**Authors:** Danila Amoroso, Paolo Barone, Silvia Picozzi

**Affiliations:** 1Consiglio Nazionale delle Ricerche CNR-SPIN, c/o Università degli Studi “G. D’Annunzio”, I-66100 Chieti, Italy; silvia.picozzi@spin.cnr.it; 2Consiglio Nazionale delle Ricerche CNR-SPIN, Area della Ricerca di Tor Vergata, Via del Fosso del Cavaliere 100, I-00133 Rome, Italy; paolo.barone@spin.cnr.it

**Keywords:** topology and magnetism, topological spin textures, magnetic interactions, atomic scale magnetic properties, magnetic nanostructures

## Abstract

The effects of competing magnetic interactions in stabilizing different spin configurations are drawing renewed attention in order to unveil emerging topological spin textures and to highlight microscopic mechanisms leading to their stabilization. The possible key role of the two-site exchange anisotropy in selecting specific helicity and vorticity of skyrmionic lattices has only recently been proposed. In this work, we explore the phase diagram of a frustrated localized magnet characterized by a two-dimensional centrosymmetric triangular lattice, focusing on the interplay between the two-ion anisotropy and the single-ion anisotropy. The effects of an external magnetic field applied perpendicularly to the magnetic layer, are also investigated. By means of Monte Carlo simulations, we find an abundance of different spin configurations, going from trivial to high-order Q skyrmionic and meronic lattices. In closer detail, we find that a dominant role is played by the two-ion over the single-ion anisotropy in determining the planar spin texture; the strength and the sign of single ion anisotropy, together with the magnitude of the magnetic field, tune the perpendicular spin components, mostly affecting the polarity (and, in turn, the topology) of the spin texture. Our analysis confirms the crucial role of the anisotropic symmetric exchange in systems with dominant short-range interactions; at the same time, we predict a rich variety of complex magnetic textures, which may arise from a fine tuning of competing anisotropic mechanisms.

## 1. Introduction

The competition between different magnetic interactions, leading to the so-called *frustration*, is a key ingredient for the stabilization of noncollinear and noncoplanar magnetic configurations. In particular, anisotropic interactions play a crucial role in the formation of exotic and topological spin textures, such as Bloch or Neél type skyrmions and anti-skyrmions, most commonly having a topological charge |Q| equal to one [1,2,3]. Nevertheless, topological spin textures go beyond conventional skyrmions, from high-*Q* (anti)skyrmionic [4,5,6,7] to halved-*Q* (anti)meronic spin textures and lattices [4,8,9,10,11,12,13]. The different types of isolated two-dimensional (2D) topological spin textures are primarily characterized by the polarity *p* and vorticity *m*, whose product, Q=m·p, defines their topology. The polarity is associated with the out-of-plane magnetization profile, when moving from the core of the topological object to its edge (to infinity in a continuous description of isolated object); the vorticity is determined by the in-plane magnetization rotation and it is also referred to as the winding number, allowing only for integer values [8,14]. Therefore, (anti)skyrmions, which are characterized by reversed magnetization directions when comparing the core with its edge, corresponding to a spin configuration wrapping a unit sphere, always display unitary values of the polarity (p=±1) and, hence, integer values of *Q*. In (anti)merons, instead, the magnetization at the edge and at the core are directed perpendicularly to each other, thus corresponding to the wrapping of only half of the sphere and to halved polarity, p=±1/2. In this case, the associated topological charges are multiples of half-integer values. Note that the sign of the topological charge *Q* changes under time reversal, reversing the *p* sign.

Conventional Bloch-type skyrmion lattices (m=1) are usually observed in chiral magnets, such as noncentrosymmetric B20-MnSi prototypical example [15,16]. The skyrmion formation is primarily driven by competing Heisenberg and Dzyaloshinskii–Moriya (DM) [17] exchange interactions, the sign of the latter determining the spins rotational direction, in turn dictated by the chiral crystal structure [18,19,20,21]. Skyrmionic lattices with various—not *a priori* determined—topologies can instead occur in geometrically frustrated lattices (such as triangular or Kagome), triggered by competing magnetic exchange interactions and assisted by dominant non-chiral interactions, such as easy-axis anisotropy [22,23,24,25,26,27,28,29], long-range dipole–dipole and/or Ruderman–Kittel–Kasuya–Yosida (RKKY) interactions [7,30,31,32,33], and thermal or quantum fluctuations [34]. In both cases, the stabilization of skyrmions is often driven by an external magnetic field perpendicular to the magnetic layer (i.e., out-of-plane). On the other hand, the formation of meronic textures is usually driven by an in-plane magnetic field and/or easy-plane anisotropy. The effect of such in-plane magnetic interactions is indeed to penalize the out-of-plane magnetization of skyrmion-like configurations. This may affect their size and, consequently, their overlap when arranged on lattices, causing a fractionalization of the topological charge and, possibly, a transition to meronic lattices [10,11,12,13,35,36].

So far, the spontaneous stabilization of skyrmionic lattices has been proposed only in itinerant magnets, characterized by long-range exchange interactions mediated by conduction electrons [1,37,38,39]. Only recently, a frustrated semiconducting 2D-magnet, NiI${}_{2}$ monolayer, was predicted to show the spontaneous formation of a stable high-*Q* skyrmionic lattice (Q=2) [40], with well defined topology and chirality determined by the anisotropic part of the short-range symmetric exchange, in the absence of DM and Zeeman interactions. In particular, the authors pointed out that the anisotropic symmetric exchange, also referred to as two-ion anisotropy or bond-dependent exchange anisotropy [41], may act as an emergent chiral interaction, thus adding frustration in the relative orientation of spins, when displaying noncoplanar principal axes, and determining the topology of the localized spin texture.

Given these premises, in this work, we elaborate on our recent findings by investigating the effects of the interplay between the two-ion anisotropy (TIA) and single-ion anisotropy (SIA) on the formation and stabilization of topological spin structures. A rich phase diagram is uncovered by means of Monte Carlo simulations, where both SIA and applied out-of-plane magnetic field trigger various transitions between different kinds of magnetic configurations, with the topological charge ranging from high-*Q* to zero values. The previously identified [40] spontaneous high-*Q* skyrmion lattice is found to extend over a significant range of SIA, confirming the robustness of the TIA-based mechanism. On the one hand, the in-plane component of all noncoplanar emerging triple-q spin states is found to be always fixed by the two-ion anisotropy, confirming its primary role in determining the rotational direction of in-plane magnetization over the SIA. On the other hand, the out-of-plane magnetization is tuned by both SIA and applied magnetic field, thus affecting mostly the polarity and, hence, the spin texture topology. It is worth to note that similar magnetic phases have been recently reported also in frustrated itinerant magnets [42], arising from the interplay of magnetic anisotropies and dominant long-range and higher-order spin interactions, such as the biquadratic exchange, mediated by conduction electrons and spin-charge coupling. As such, the study reported in Ref. [42] for itinerant systems is different from the one presented here, where the focus is on short-range interactions between localized magnetic moments.

## 2. Materials and Methods

The starting point of our analysis is the classical spin Hamiltonian (given in the following Equation (Equation 1)) describing the magnetic properties of the semiconducting and centrosymmetric NiI${}_{2}$ monolayer [40]. NiI${}_{2}$ belongs to the family of transition-metal-based (*M*) van der Waals materials [43,44,45,46,47,48,49,50], comprising layers where magnetic cations form a 2D triangular lattice, thus displaying *geometrical frustration* [51,52]. Our previous first-principles based investigation [40] on NiI${}_{2}$ monolayer revealed that its magnetic properties are ruled by:(i)strong magnetic frustration arising from competing nearest-neighbor ($J^{1iso}$) ferromagnetic (FM) and third-nearest neighbor ($J^{3iso}$) anti-ferromagnetic (AFM) exchange interactions;(ii)very weak easy-plane single-ion anisotropy against highly anisotropic symmetric exchange, both driven by spin-orbit coupling (SOC).

Monte Carlo (MC) simulations have shown that the combination of these two properties result in the spontaneous stabilization of a topological spin structure with well defined topology and chirality.

In closer detail, the model Hamiltonian for classical spins of unit length $s_{i}=1$ is defined as:(1)$$H=\frac{1}{2}\sum_{i\neq j}s_{i}J_{ij}s_{j}+s_{i}A_{i}s_{i}$$
where **J**${}_{ij}$ and **A**${}_{i}$ are tensors describing the exchange interaction and single-ion anisotropy, respectively [53]. It is convenient to decompose the exchange coupling tensor into an isotropic part $J_{ij}^{iso}=\frac{1}{3}\mathrm{Tr}J_{ij}$ and an anisotropic symmetric part $J_{ij}^{S}=\frac{1}{2}\left(J_{ij}+J_{ij}^{T}\right)-J_{ij}^{iso}I$, herein also referred to as two-ion anisotropy. Due to the inversion symmetry of the lattice (with $D_{3d}$ point group), the DM interaction, which corresponds to the antisymmetric exchange term $J_{ij}^{A}=\frac{1}{2}\left(J_{ij}-J_{ij}^{T}\right)$, is identically zero. Reference magnetic parameters for NiI${}_{2}$ monolayer, evaluated from density-functional theory (DFT) calculations, are given in Table 1. The interactions were estimated by means of the four-state energy mapping method [53], performing PBE+U+SOC calculations [54,55] (*U* = 1.8 eV, *J* = 0.8 eV) via the VASP code [56,57]. We used a 5×4×1 supercell to estimate the SIA and first nearest-neighbor interactions, and a 6×3×1 supercell for the estimate of the third nearest-neighbor interaction. Supercells were built from the periodic repetition of the optimized NiI${}_{2}$ monolayer unit cell, with the lattice parameter of about 3.96 Å and a vacuum distance of ≃17.5 Å between periodic copies of the free-standing layer along the *c* axis. The matrix elements of the two-ion anisotropy tensor ($J^{S}$ or TIA) between nearest-neighbour spins, are expressed in a local cartesian {x,y,z} basis, where *x* is parallel to the Ni-Ni bonding vector, and to the lattice vector **a**. Further details can be found in the Methods section of Ref. [40].

Taking into account such terms, the phase diagram of the magnetic system, arising when tuning the single-ion anisotropy and applied out-of-plane magnetic field, has been studied within a Monte Carlo approach. In particular, MC calculations were performed using a standard Metropolis algorithm on L×L triangular supercells with periodic boundary conditions. Starting from high temperature (*T*), at each simulated *T*, we used 10${}^{5}$ MC steps for thermalization and 5 × 10${}^{5}$ MC steps for statistical averaging. The lateral size of the simulation supercells was chosen as $L=nL_{m.u.c.}$, where *n* is an integer and $L_{m.u.c.}$ is the lateral size of the magnetic unit cell, in units of the lattice constant $a_{0}$, needed to accommodate the lowest-energy noncollinear helimagnetic spin configurations. Accordingly, we estimated $L_{m.u.c.}$ as 1/q, where *q* is the length of the propagation vector **q** minimizing the isotropic exchange interaction in the momentum space, *J*(**q**). Using the magnetic parameters listed in Table 1, the propagation vector for the isotropic model is given by $q=2\cos^{-1}\left[\left(1+\sqrt{1-2J^{1iso}/J^{3iso}}\right)/4\right]$ [26,33], resulting in $L_{m.u.c.}≃8$. The results presented are thus obtained by means of calculations performed on supercells with lateral size $L=3L_{m.u.c.}=24$.

Further insights on the magnetic configurations are obtained by evaluating the spin structure factor:(2)$$\begin{array}{ccc} S(q) & = & \frac{1}{N}\;\sum_{\alpha =x,y,z}\;\langle {\left|\sum_{i}s_{i,\alpha }\;e^{-iq\cdot r_{i}}\right|}^{2}\rangle \end{array}$$
where $r_{i}$ denotes the position of spin $s_{i}$ and $N=L^{2}$ is the total number of spins in the supercell used for the MC simulations. The bracket notation is used to denote the statistical average over the MC configurations. The spin structure factor provides direct information on the direction and size of the propagation vectors. The topological nature of the multiple-q phases has been assessed by evaluating the topological charge (or skyrmion number) of the lattice spin field of each supercell as $\langle Q\rangle =\langle \sum_{i}\Omega_{i}\rangle$; $\Omega_{i}$ is calculated for each triangular plaquette as [58]
(3)$$\begin{array}{c} \tan\left(\frac{1}{2}\Omega_{i}\right)=\frac{s_{1}\cdot s_{2}\times s_{3}}{1+s_{1}\cdot s_{2}+s_{1}\cdot s_{3}+s_{2}\cdot s_{3}} \end{array}$$

In the following, starting from the exchange interactions estimated in the prototypical monolayer NiI${}_{2}$ as representative example of highly frustrated semiconducting 2D triangular lattice systems, we explore the magnetic phase diagram as a function of the strength and direction of the SIA as well as of an applied out-of-plane magnetic field $B_{z}$.

## 3. Results

In Figure 1, we show the low temperature (T=1 K) magnetic phase diagram in the SIA-field ($A_{zz}-B_{z}$) plane, obtained by fixing the exchange coupling interactions to the reference values reported in Table 1, and varying the value and direction of the single-ion anisotropy; for each $A_{zz}$ value, an increasing magnetic field ($B_{z}$) was also applied along the perpendicular direction of the monolayer triangular lattice. As shown in Figure 1a, for each combination of exchange coupling and $A_{zz}$ parameters, there are field-induced topological phase transitions between states with different topological charge *Q*, as listed below, approximately, per magnetic unit cell (m.u.c.):
$-0.75≲A_{zz}/{\left|J\right|}^{1iso}≲-0.30$${\left|Q\right|}_{m.u.c}=0\to 6\to 3\to 0$$-0.30≲A_{zz}/{\left|J\right|}^{1iso}≲0.30$${\left|Q\right|}_{m.u.c}=6\to 3\to 0$$0.30≲A_{zz}/{\left|J\right|}^{1iso}≲0.50$${\left|Q\right|}_{m.u.c}=6\to 3\to 1.5\to 0$$0.50≲A_{zz}/{\left|J\right|}^{1iso}≲0.65$${\left|Q\right|}_{m.u.c}=0\to 6\to 3\to 0$$0.65≲A_{zz}/{\left|J\right|}^{1iso}≲0.75$${\left|Q\right|}_{m.u.c}=0\to 3\to 0$

As schematically represented in Figure 1c, the phase diagram consists of various trivial and topologically equivalent phases—i.e., displaying the same $Q_{m.u.c}$—but characterized by different kinds of spin configurations. As shown in Figure 1b, for different realizations of the easy-axis and easy-plane anisotropy, field-induced phase transitions are accompanied by discontinuities of the out-of-plane magnetization $M_{z}$. Within the considered range of applied $B_{z}$, the magnetization saturation $M_{z}=1$, corresponding to all spins aligned parallel to the magnetic field, is reached only in the case of easy-axis anisotropy and for small easy-plane anisotropy; indeed, the latter penalizes the out-of-plane component of the spins. The in-plane magnetization is zero in all cases.

In the following, we describe and analyze the various anisotropy-field induced magnetic orders (Figure 2, Figure 3, Figure 4, Figure 5 and Figure 6), also providing visual details of the out-of-plane and in-plane components of spins for the various phases (Figure 7 and Figure 8.)

### 3.1. Topologically Trivial Spin Orderings

We start our analysis by discussing the topologically trivial spin configurations found in the strong-field region of the phase diagram, and in the weak-field region for strong SIA (both easy-axis and easy-plane). In Figure 2, we show both spin configurations and topological charge densities ($\Omega_{i}$) alongside the associated spin structure factor of such phases, labeled as (A), (B) and (H) in the phase diagram of Figure 1c.

For weak fields, two helical single-**q** states (A and B phases) appear for strong easy-axis and easy-plane SIA, whose spin configurations are shown in Figure 2a,d, respectively. No finite topological charge appears in such phases, as clearly shown in Figure 2b,e. From Figure 2c,f, the propagation vector is $q_{1}=\left(-2\delta ,\delta \right)$, with q/2=δ≃1/8 in reduced coordinates, being mostly determined by frustrated isotropic exchange. Due to the three-fold rotational symmetry of the triangular lattice, the single-q helices propagating along the *x*-axis with $q_{1}$ are equivalent (and, hence, energetically degenerate) to helices propagating along the symmetry-equivalent directions rotated by $+120^{\circ }$, with $q_{2}=\left(\delta ,-2\delta \right)$, and $-120^{\circ }$, with $q_{3}=\left(\delta ,\delta \right)$. The plane of spins rotation is selected by both SIA and TIA, which thus determine the nature of the helical states. The inspection of both spin structures and spin structure factors reveals that: *(i)* phase (A) consists of a proper-screw spiral propagating along the Cartesian *x*-axis, i.e., along the Ni-Ni bond direction, with spins rotating in the perpendicular yz-plane; *(ii)* phase (B) is a tilted cycloid, i.e., a helix where spins rotate in a plane containing the propagation vector (parallel to *x*) but tilted around it, causing the spins to acquire a non-zero *z*-component highlighted by the peaks of $S_{z}\left(q\right)$ (cfr Figure 2f). The modeling of the single-q proper-screw and cycloidal helices is presented in Appendix A.

The combination of the three spirals defines a triple-q state, that is the case for the H-phase depicted in Figure 2g. This is a trivial noncollinear spin configuration, but characterized by local nonzero scalar chirality, as evident from the colormap of the topological charge density in Figure 2. Indeed, it is possible to recognize a hexagonal lattice formed by six vortices (*m* = +1) with chirality opposite to the surrounded anti-bi-vortex core (m=−2). This state occupies a wide region of the phase diagram for high values of applied magnetic field, before evolving into a pure ferromagnetic state [phase (I), Figure 2j]. In addition to the weak peaks at $q_{1}$, $q_{2}$, and $q_{3}$ (Figure 2i), this H-phase displays an out-of-plane ferromagnetic component, as shown by the relevant peak in the S(q) at the Γ-point (q=0), which is also reflected in the remarkable out-of-plane component of the magnetization, $0.5≲M_{z}<1$ (Figure 1b, as well as in the $s_{z}$ profile depicted in Figure 8m). Noteworthy, such triple-q trivial state is largely favored by the easy-plane anisotropy, which indeed competes with $B_{z}$, penalizing the out-of-plane spin component; indeed, for $A_{zz}/{\left|J\right|}^{1iso}>0.15$ ($A_{zz}/\left|J_{yz}\right|>0.75$), the FM state is never achieved within the wide range of explored $B_{z}$. Conversely, the easy-axis anisotropy sustains the $B_{z}$ action, favoring the out-of-plane spin orientation; accordingly, the region of the phase diagram occupied by the H-phase reduces, until it disappears for $A_{zz}/{\left|J\right|}^{1iso}≲-0.75$ (Figure 1c).

### 3.2. Topological Spin Orderings

In this section, we turn our attention to the various topological spin configurations which can be realized by tuning the competition between the single-ion anisotropy and the applied field. The related spin textures, topological charge densities and spin structure factors are shown in Figure 3, Figure 4 and Figure 5, respectively. All topological lattices are triple-**q** states, with **q**${}_{1}=\left(-2\delta ,\delta \right)$, **q**${}_{2}=\left(\delta ,-2\delta \right)$, **q**${}_{3}=\left(\delta ,\delta \right)$ and δ≃1/8 [Figure 3c,f, Figure 4c,f and Figure 5c], characterized by atomic scale skyrmionic or meronic structures composed by nano-sized topological objects with a radius counting ≃4 spins, and thus a diameter of few units of the lattice parameter $a_{0}$.

#### 3.2.1. Topological Lattice with ${\left|Q\right|}_{m.u.c}=6$

In Figure 3a,d, we show the high-*Q* topological lattices labelled C- and D- phases, respectively, which occupy the region of the phase diagram for zero and low magnetic field; in particular, the C-phase is the spontaneous topological phase reported in [40], and stabilized by the exchange magnetic couplings given in Table 1.

In closer detail, the topological lattice of the C-phase ranges over $-0.30≲A_{zz}/{\left|J\right|}^{1iso}≲+0.50$ and $-1.50≲A_{zz}/\left|J_{yz}\right|≲+2.50$ for zero field, whereas it slightly extends over the easy-plane anisotropy part of the phase diagram upon low $B_{z}$, as shown in Figure 1a,c. It is a triple-**q** state characterized by a hexagonal lattice formed by six vortices (V, m=+1) with downward central spin, hosting at the center of each hexagon an anti-bi-vortex (A2V, m=−2) with upward core, as shown in Figure 3a. Such spin texture defines a homogeneous topological charge density, as depicted in Figure 3b, hence uniform scalar chirality, and gives rise to a topological charge of six per magnetic unit cell. This topological lattice can be regarded as the periodic repetition of topological objects, consisting of two vortices and one anti-bi-vortex (schematically depicted in Figure 7g), all contributing to defining a global |Q|=2. As evident from the $s_{z}$ profile reported in Figure 7a, the central A2V does not display a uniform, unitary polarity along its surrounding perimeter (highlighted as a purple circle traced in Figure 7b): the upward spin of the core ($s_{z}=+1$) is not reversed downward ($s_{z}=-1$) at all points of its finite edge. Therefore, it brings a fractionalized topological charge $Q_{A2V}=-2\cdot p_{A2V}$, with $0<p_{A2V}<+1$. The missing *Q* fraction is carried by the two vortices, which, similarly, bring a fractionalized charge $Q_{V}=+1\cdot p_{V}$, with $-1<p_{V}<0$. Accordingly, $\left|Q|=\right|Q_{A2V}+2Q_{V}|=2$. The minimal magnetic cell accommodating this topological lattice consists of the composition of three repeated objects; hence ${\left|Q\right|}_{m.u.c}=6$. The topological lattice can therefore be interpreted as a fractionalized anti-bi-skyrmion (A2SK) lattice, where the fractionalization of the topological charge can be ascribed to the A2SKs close packing, leading to the incomplete spin wrapping highlighted in Figure 7 [10]. A similar realization of a fractionalized skyrmion lattice has been recently reported in MnSc${}_{2}$S${}_{4}$, where an applied field has been experimentally shown to stabilize a lattice of fractionalized Bloch-type skyrmions and incipient merons [59].

By increasing the easy-axis anisotropy, i.e., for $-0.75≲A_{zz}/{\left|J\right|}^{1iso}≲-0.45$ and $-3.75≲A_{zz}/\left|J_{yz}\right|≲-2.25$, keeping low values of the applied magnetic field, the magnetic phase transforms into the D-phase shown in Figure 3d. Despite the weak FM component seen in the spin structure factor (Figure 3f), which is compatible with both applied field and enhanced easy-axis anisotropy, the spin configuration approaches an ideal triangular lattice of anti-bi-skyrmions: each anti-bi-vortex is surrounded by a magnetic background, where almost all spins are fully reversed with respect to the core of the A2V along the radial directions, as depicted in Figure 8a,b. Nevertheless, the proximity of the small sized A2SKs still produces a weak overlap between these topological objects, in turn, causing the formation of residual vortices at the center of the triangles formed by three nearest-neighbor A2SKs [10]; therefore, these anti-bi-skyrmions carry a fractionalized topological charge, that is $1.5≲|Q_{A2SK}|<2$ (or equivalently $4.5≲{\left|Q\right|}_{m.u.c}<6$ considering three anti-bi-skyrmions per magnetic unit cell). Accordingly, the topological charge density map is no longer homogeneous, rather the highest intensities are localized around the anti-bi-vortex core (Figure 3e). We finally notice that the cores of each A2SK is found to align anti-parallel to the applied field direction. The spin texture shown in Figure 3d has been obtained by applying a negative $B_{z}$ to ease the comparison with the C-phase displayed in Figure 3a; we verified that a reversal of the magnetic field direction systematically causes a reversal of the A2SK-core magnetization and an alignment of the magnetic background with the field.

#### 3.2.2. Topological Lattice with ${\left|Q\right|}_{m.u.c}=3$

In Figure 4a,d, we show the topologically equivalent ${\left|Q\right|}_{m.u.c}=3$ lattices, labelled E- and F-phases, respectively, which occupy a wide portion of the phase diagram in an intermediate range of applied magnetic field $B_{z}$ (Figure 1c). Despite carrying the same topological charge, their spin textures are markedly different.

As shown in Figure 4a, the E-phase consists of a triangular skyrmion lattice, with vortices and anti-bi-vortices appearing at the intersection of clearly visible Bloch-like skyrmions, namely, at the center of the triangles that build up the skyrmion lattice. Such phase can be seen as a direct transition of the stable C-phase induced by intermediate values of the applied perpendicular field: upon $B_{z}$, spins of the anti-bi-vortex core remain parallel to the field, spins of one of the two vortices also tend to align with the magnetic field, while spins at the core of the other vortex remain anti-parallel. Such a change in the $s_{z}$ spins component in one of the two vortices affects the spin scalar chirality, thus leading to the topological transition from ${\left|Q\right|}_{m.u.c}=6$ in the C-phase to ${\left|Q\right|}_{m.u.c}=3$ in the E-phase. In the latter phase, the topological charge density map shows in fact negligible contributions from the A2V, while giving rise to an opposite sign of $\Omega_{i}$ for the upward and downward vortices. The highest contribution to the topological charge arises from the vortices whose core-magnetization remains antiparallel to the applied field, thus defining the Bloch-type skyrmion lattice. Indeed, the corresponding $s_{z}$ profile, shown in Figure 8g,h, testifies for an almost full reversal of the out-of-plane spin component, i.e., polar angle variation of about $180^{\circ }$, when moving from the core ($s_{z}=-1$) to the edge ($s_{z}=+1$) of the Bloch-type vortex.

Such skyrmion lattice is found to be robust with respect to easy-axis anisotropy. On the other hand, a strong enough single-ion easy-plane anisotropy ($A_{zz}>>0$) induces its transformation into the F-phase, observed in a narrow region of the investigated phase diagram (see Figure 1c): the out-of-plane component of magnetization antiparallel to the applied field is here further suppressed by the strong easy-plane SIA. In particular, the analysis of the spin texture shown in Figure 4d, combined with the $s_{z}$ profile (Figure 8d,e) and the topological charge, indicates that the F-phase is a lattice of anti-bi-merons (A2M): the A2V of the C-phase preserves the upward core, with spins parallel to $B_{z}$, while the surrounding vortices preserve only the planar spins components. Indeed, the polar angle changes by $90^{\circ }$ when moving along the radial direction from the core to the edge, i.e., p=+0.5, as from the $s_{z}$ profile in Figure 8d. As a consequence, the topological charge of the F-phase is halved with respect to that of the C-phase, that is ${\left|Q\right|}_{m.u.c}=6\to 3$. To the best of our knowledge, this is the first time that such kind of A2M texture is reported.

#### 3.2.3. Topological Lattice with ${\left|Q\right|}_{m.u.c}=1.5$

As a last relevant topological phase composing the phase diagram, we show in Figure 5a a meronic lattice with ${\left|Q\right|}_{m.u.c}=1.5$, which can be regarded as an intermediate spin configuration during the evolution of the SK-lattice of the E-phase into the trivial triple-(**q**) ferromagnetic H-phase upon $B_{z}$ (Figure 1c). The competition of the intermediate-strong easy-plane anisotropy with the intermediate-strong applied field causes the transformation of the Bloch-like skyrmion into a meron with halved topological charge. As shown by the spin texture in Figure 5a and the $s_{z}$ profile of the vortex core in Figure 8j,k, the latter loses the spin component along the *z*-direction, while it remains aligned to the field in the surrounding vortices and anti-bi-vortices, which occupy alternatively the center of the triangles formed by the merons: spins at the edge of the topological vortex are directed perpendicularly with respect to its core, halving the topological charge of the Bloch-like skyrmion lattice of the E-phase, ${\left|Q\right|}_{m.u.c}=3\to 1.5$.

### 3.3. Field-Induced Phase Transition from Two-Ion Anisotropy Tuning

In Figure 6, we report the evolution of the topological charge per magnetic unit cell ${\left|Q\right|}_{m.u.c}$ as a function of increasing $B_{z}$, and for different values of the off-diagonal $J_{yz}$ exchange coupling term, which is treated here as a measure of the non-coplanarity induced by the two-ion anisotropy. SIA is kept fixed to the NiI${}_{2}$ reference value (Table 1); in fact, from the phase diagram previously discussed (Figure 1), we found that the SIA is basically ineffective against the anisotropic symmetric exchange interaction for a wide range of values (i.e., $-1.50≲A_{zz}/\left|J_{yz}\right|≲+2.50$ and $-0.30≲A_{zz}/{\left|J\right|}^{1iso}≲+0.50$).

For $J_{yz}>0.14\;J^{1iso}$ (the reference value is $J_{yz}≃0.20\;J^{1iso}$), the topological lattice with ${\left|Q\right|}_{m.u.c}=6$, represented by the C-phase (Figure 3a), is the lowest energy spin configuration stabilized by the considered magnetic interactions. In particular, in line with the results shown in Figure 1a, two sharp topological phase-transitions, signalled by the abrupt change of the total topological charge, are induced under applied magnetic fields: ${\left|Q\right|}_{m.u.c}=6\to 3\to 0$. The ${\left|Q\right|}_{m.u.c}=3$-state obtained under intermediate $B_{z}$ is the Bloch-type skyrmion lattice of the E-phase; for strong $B_{z}$, the transition to the trivial ferromagnetic I-phase takes place passing through the intermediate H-phase, characterized by the topologically trivial triple-**q** state (Figure 2j,g, respectively).

For $J_{yz}\leq 0.14J^{1iso}$, two topological phase transitions still take place, but the stable phase at zero and weak magnetic field is now a trivial single-**q** state; $|Q_{m.u.c}|=0\to 3\to 0$ as a function of $B_{z}$. In particular, such single-**q** state is found to be a proper-screw spin-spiral. The magnetic phases induced by the applied fields are closely related to those observed when $J_{yz}>0.14J^{1iso}$. Therefore, a Bloch skyrmion lattice can still be stabilized by a magnetic field applied on a topologically trivial helical state, when the non-collinearity brought in by the two-ion anisotropy is not sufficiently strong to drive a spontaneous high-*Q* topological lattice.

## 4. Discussion

In the previous section, we have described the rich anisotropy-field phase diagram (Figure 1) obtained by tuning strength and direction of the single-ion anisotropy (from easy-plane to easy-axis) as well as upon increasing magnetic field, while keeping fixed the strength of both the isotropic and anisotropic symmetric exchange interactions describing the short-range magnetic interactions of the spin-spin Hamiltonian (Equation 1) for a centrosymmetric triangular lattice. We found that, considering the magnetic interactions reported in Table 1, the spontaneous C-phase, consisting of a hexagonal lattice of vortices (m=+1) and anti-bi-vortices (m=−2) with a total topological charge of six per magnetic unit cell, is a persistent thermodynamically-stable phase in a wide portion of the SIA-field phase diagram; it is, to a wide extent, independent on the easy-plane or easy-axis character of the single-ion anisotropy. Our analysis also unveils different field-induced topological phase transitions that can be realized for different values of the SIA coupling constant.

Nevertheless, it is important to notice that all the different emerging triple-(**q**) states, ranging from topological (phases C, D, E, F, G) to trivial (H-phase) spin configurations, exhibit a common planar spin texture. To better appreciate this feature, we show in Figure 7c and Figure 8c,f,i,l,o the color-gradient plots of the in-plane spin components highlighting the azimuthal angle $\theta \in [0^{\circ },360^{\circ })$, defined by the $s_{x}$ and $s_{y}$ components of the spin vector s, and which is related to the in-plane spins rotational direction.

Such specific in-plane orientation of spins can be traced back to the TIA, which tends to orient the spins along given noncollinear and noncoplanar orientations in space, defining both the helicity and vorticity of the spin pattern and thus behaving as an emergent chiral interaction. Indeed, as discussed in Ref. [40], the principal magnetic axes per bond of the $J^{S}$ tensor are not all parallel either to the lattice vectors or with the principal axes of neighbouring bonds, as shown in Figure 7d. This introduces noncoplanar components in the spin-spin interaction and, thus, frustration in the relative orientation of the spins, which ultimately arises from the non-coplanarity of the spin-ligand M−X plaquettes mediating the exchange interactions in the spin triangular lattice. In particular, the in-plane projection of the principal axis directed along the ligand X−X direction [indicated by red arrows in Figure 7d,e] fixes the in-plane orientation of the nearest-neighbour magnetic moments giving rise to the A2V spin-pattern [Figure 7e,f]. The accommodation of the anti-bi-vortices in the spin lattice, which tend to overlap because of the short-period modulation due to the isotropic magnetic frustration ($J^{3iso}\;vs\;J^{1iso}$), causes the emergence of the surrounding vortices.

With the given strong anisotropic symmetric exchange interaction considered in the present case, the two-ion anisotropy dominates over the single-ion anisotropy and external magnetic field: SIA and $B_{z}$ tune the out-of-plane spins component, modifying the polarity *p*, and thus the final spins scalar chirality and related topology (*Q*) of the spin configuration; the planar spin texture results always fixed by the TIA. All the triple-(**q**) phases can be in fact seen as a transformation of the initial C-phase via major modifications of the $s_{z}$ component of spins; the in-plane directions of the spins remaining unchanged. Moreover, even though single-q states are found to be energetically more stable than triple-q ones when $A_{zz}≳2.5\left|J_{yz}\right|$ and $A_{zz}≲-1.5\left|J_{yz}\right|$ (or also when $J_{yz}\leq 0.14J^{1iso}$), an applied out-of-plane magnetic field can still induce a transition to a topological skyrmion lattice, whose in-plane spin configuration appears to be determined by the two-ion anisotropy.

Beyond the fundamental interest of our findings, which identifies the distinct role played by different competing magnetic interactions, the rich anisotropy-field phase diagram calls for further studies. In particular, the investigation of magnetic properties in other two-dimensional van der Waals materials and their possible interplay with dielectric properties, as also pointed out by recent experimental achievements [60,61], open the path to new low-dimensional magnetoelectrics and/or multiferroics.

## 5. Conclusions

In this work, we have theoretically investigated the effects of competing single-ion and two-ion anisotropies in a triangular lattice with strong magnetic frustration, as occurring in monolayers of van der Waals nickel dihalides. By means of Monte Carlo calculations, we analyzed the parameter space spanned by SIA and applied $B_{z}$ field for a spin-lattice model whose parameters have been estimated for a monolayer of prototypical semiconducting NiI${}_{2}$. Our analysis has revealed a rich phase diagram comprising different magnetic phases, from topologically trivial single-(**q**) and triple-(**q**) states to topological triple-(**q**) states. The strong magnetic frustration, arising from the competing isotropic nearest-neighbour FM and third nearest-neighbour AFM exchange interactions, promotes the onset of short-period helimagnetic configurations, whereas the strong two-ion anisotropy combined with the geometrical frustration of the underlying triangular lattice favours the stabilization of triple-q states. At zero magnetic field, these result in a lattice of vortices and anti-(bi)-vortices carrying a total topological charge per magnetic unit cell ${\left|Q\right|}_{m.u.c}=6$, which is robust within a wide range of single-ion anisotropy strength. Such topological lattice can be interpreted as a fractionalized anti-bi-skyrmion lattice where each anti-bi-skyrmion is surrounded by six vortices arising from the overlap with neighbouring anti-bi-skyrmions, causing a fractionalization of the topological charge of individual anti-bi-vortices. Both the single-ion anisotropy and the applied field act primarily on the out-of-plane component of the spin configurations, hence on the polarity *p* of the vortical states, modulating both the size of anti-bi-vortices and the fractionalization of their charges. As a general trend, on the one hand, easy-axis anisotropy is found to increase the localization of anti-bi-vortices, thus reducing the “spilling” of topological charge to surrounding vortices; on the other hand, a strong easy-plane anisotropy, when combined with applied field, would eventually favour a crossover to meronic lattices, i.e., lattices formed by topological objects with half-integer polarity. Nevertheless, an out-of-plane magnetic field, sustained by the strong anisotropic exchange, is always found to trigger a topological transition to a Bloch-like skyrmion lattice, when applied either on a single- or a triple-q state. Interestingly, the in-plane components of the spin texture are found to be extremely robust across almost the whole phase space explored in this work, with exceptions only for extreme values of SIA and fields.

In conclusion, the different topological phases forming the SIA-field phase diagram can be understood by a change of their polarity, which is directly tuned by both the single-ion anisotropy and the applied field, whereas their vorticity can be always traced back to the two-ion anisotropy on the triangular lattice, which is ultimately responsible for the promotion of both topological and trivial triple-q states. Therefore, our study corroborates the potential role of frustrated anisotropic symmetric exchange in defining the vorticity of noncollinear and noncoplanar spin configurations, leading to various possible topological lattices in 2D semiconducting magnets.

## Figures and Tables

**Figure 1 nanomaterials-11-01873-f001:**
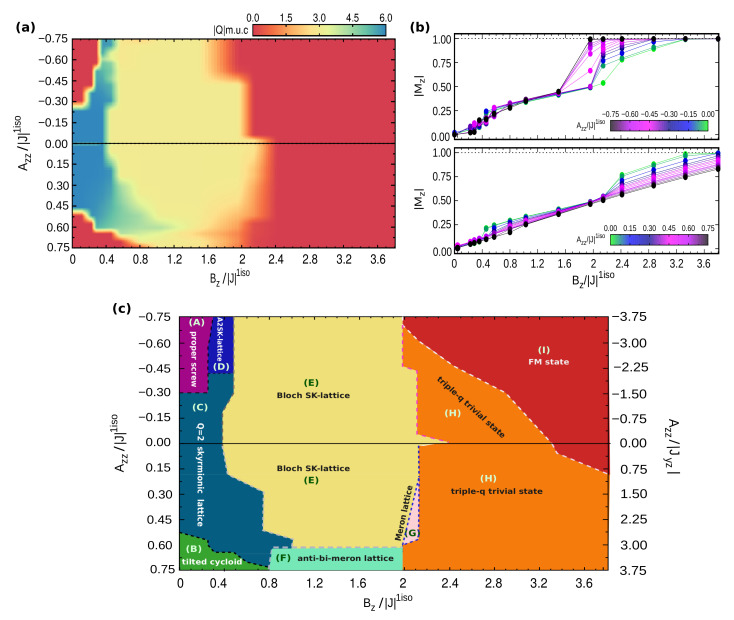
Single-ion anisotropy and field induced phase transitions. (**a**) Low temperature (T=1 K) phase diagram in the SIA-magnetic field $A_{zz}-B_{z}$ parameters space; both $A_{zz}$ and $B_{z}$ are expressed in units of the nearest neighbour isotropic exchange interaction ${\left|J\right|}^{1iso}$. Color-map indicates the absolute value of the topological charge per magnetic unit cell ${\left|Q\right|}_{m.u.c.}$. (**b**) Evolution of the out-of-plane magnetization $M_{z}$ as a function of the magnetic field $B_{z}$ for increasing SIA, easy-axis ($A_{zz}/{\left|J\right|}^{1iso}<0$, top) and easy-plane ($A_{zz}/{\left|J\right|}^{1iso}>0$, bottom), respectively. (**c**) Schematic $A_{zz}-B_{z}$ phase diagram identifying regions occupied by the different spin configurations, labelled with capital letters. Phase boundaries must be regarded only as semi-quantitative, as the field-induced topological phase transitions go through almost continuous transformations of the spin texture and metastable spin states with fractionalized topological charge, preventing the identification of accurate and sharp phase boundaries. On the right-side of the *y*-axis, $A_{zz}/\left|J_{yz}\right|$ ratio is also reported. The horizontal solid line separates the easy-axis (above) and easy-plane (below) SIA regions.

**Figure 2 nanomaterials-11-01873-f002:**
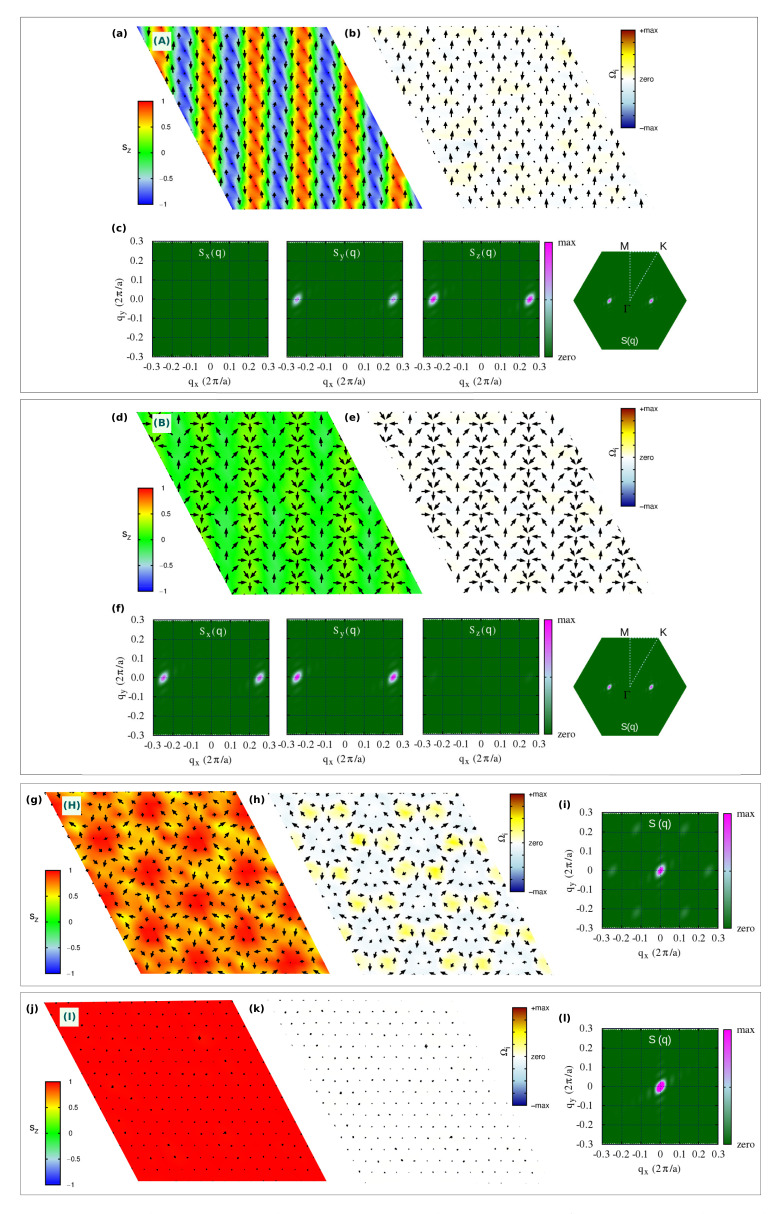
Trivial spin states with $Q_{m.u.c}=0$. Snapshots at T=1 K from Monte-Carlo simulations of real-space spin configurations (**a**,**d**,**g**,**j**) and corresponding topological charge densities $\Omega_{i}$ (**b**,**e**,**h**,**k**), in sequence: black arrows represent in-plane $\left\{s_{x},s_{y}\right\}$ components of spins; colormap indicates the out-of-plane $s_{z}$ spins component in the first snapshot and $\Omega_{i}$ in the second one. The associated spin structure factor *S*(**q**) is also shown (**c**,**f**,**i**,**l**); its decomposition in the Cartesian components $S_{x}$(**q**), $S_{y}$(**q**) and $S_{z}$(**q**) is shown for the spin spirals configurations (first and second panels). Capital letters on each spin configuration refer to the labels used in the phase diagram Figure 1c to name each spin configuration: A-phase refers to a single-(**q**) spiral (proper screw), B-phase refers to a single-(**q**) spiral (tilted cycloid), H-phase refers to a triple-(**q**) trivial state with FM component at Γ-(**q**=0), I-phase refers to a FM state.

**Figure 3 nanomaterials-11-01873-f003:**
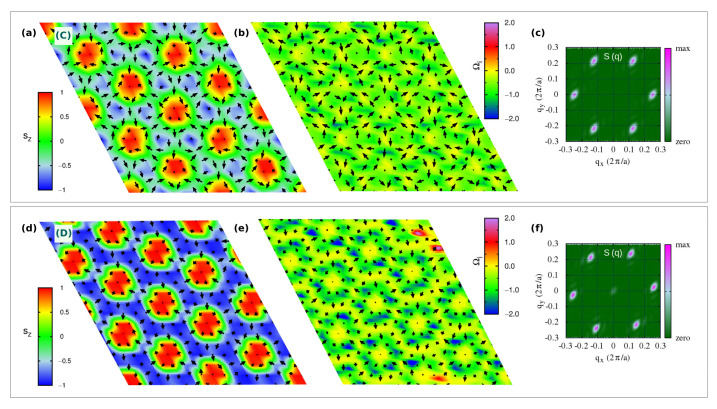
Topological lattices with ${\left|Q\right|}_{m.u.c}=6$. Snapshots at T=1 K from Monte-Carlo simulations of real-space spin configurations (**a**,**d**) and corresponding topological charge densities $\Omega_{i}$ (**b**,**e**), in sequence: black arrows represent in-plane $\left\{s_{x},s_{y}\right\}$ components of spins; colormap indicates the out-of-plane $s_{z}$ spins component in the first snapshot and $\Omega_{i}$ in the second one. The associated spin structure factor *S*(**q**) is also shown (**c**,**f**). Capital letters on each spin configuration refer to the labels used in the phase diagram Figure 1c to name each spin configuration: C-phase refers to a triple-(**q**) state, composed by periodic repetition of two vortices (V, m=+1) and an anti-bi-vortex (A2V, m=−2) with total Q=2; D-phase refers to a triple-(**q**) state, consisting of a quasi-ideal anti-bi-skyrmions (A2SK) lattice.

**Figure 4 nanomaterials-11-01873-f004:**
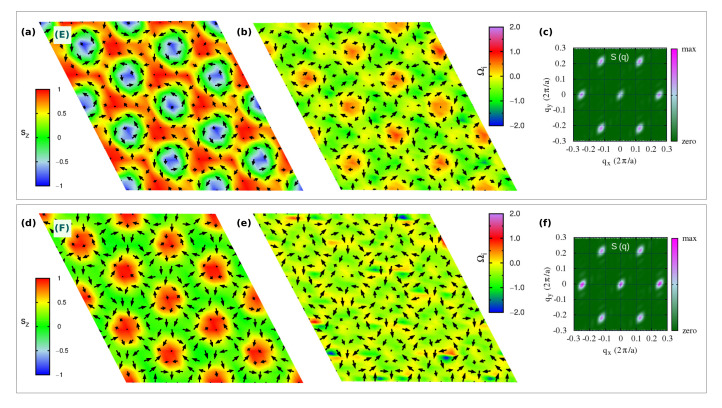
Topological lattices with ${\left|Q\right|}_{m.u.c}=3$. Similarly to Figure 3, snapshots at T=1 K of real-space spin configurations (**a**,**d**) and topological charge densities $\Omega_{i}$ (**b**,**e**), for the field-induced triple-(**q**) states: E-phase, consisting of a Bloch-type skyrmions (SK) lattice; F-phase consisting of a anti-bi-merons (A2M) lattice. The peak at q=0 in the *S*(**q**) reflects the ferromagnetic component induced by the applied $B_{z}$ (**c**,**f**).

**Figure 5 nanomaterials-11-01873-f005:**
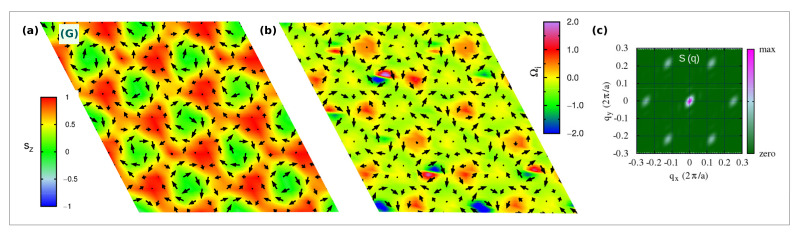
Topological lattice with ${\left|Q\right|}_{m.u.c}=1.5$. Similarly to Figure 3, snapshots at T=1 K of the real-space spin configuration (**a**) and topological charge density $\Omega_{i}$ (**b**), for the G-phase, i.e., a triple-(**q**) state, consisting of a lattice of merons from the action of large easy-plane anisotropy and $B_{z}$, as from the q=0-peak in the *S*(**q**) (**c**), on the SK-lattice of the E-phase.

**Figure 6 nanomaterials-11-01873-f006:**
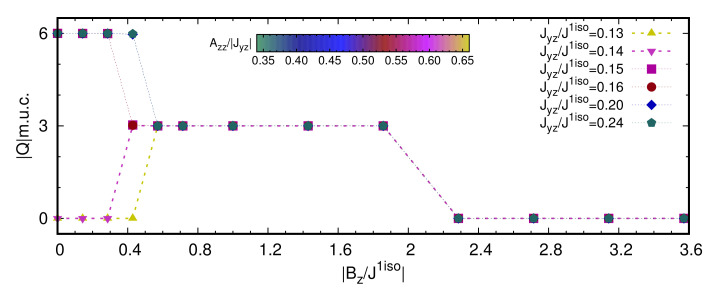
Evolution of |Q| per magnetic unit cell ($L_{m.u.c.}=8a_{0}$), as a function of the magnetic field $B_{z}$ for different off-diagonal terms, while keeping fixed diagonal exchange parameters and easy-plane anisotropy to values reported in Table 1. The color gradient of the solid points evolves with the $A_{zz}/\left|J_{yz}\right|$ ratio. At zero and low $B_{z}$ field, the system stabilizes a proper screw like the A-phase for $J_{yz}/J^{1iso}\leq 0.14$, and the topological lattice of the C-phase for $J_{yz}/J^{1iso}>0.14$. At intermediate $B_{z}$, the system undergoes a topological phase transition to a Bloch-type skyrmion lattice, E-phase. At high $B_{z}$, the transition to the ferromagnetic state, I-phase, takes place passing trough the trivial triple-q state, H-phase.

**Figure 7 nanomaterials-11-01873-f007:**
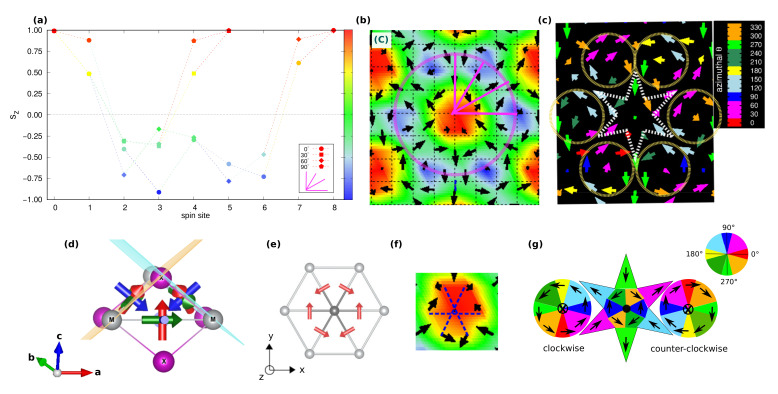
Profile of the out-of-plane component of spins and in-plane spin texture of the C-phase. (**a**) $s_{z}$ component of spins for each magnetic site from the MC simulations, when moving from the upward ($s_{z}=+1$) core (spin site n. 0) of the topological object (A2V) along the radial directions ($0^{\circ },\;30^{\circ },\;60^{\circ }\;\mathrm{and}\;90^{\circ }$, purple lines) toward the next-neighbor A2V-cores. This defines the $s_{z}$ profile, which can be regarded to as a discretized polarity *p* associated with the anti-bi-vortex, when moving from its core to its finite edge, defined by the distance with respect to the nearest-neighbor downward cores ($s_{z}=-1$) of the surrounding vortices; the A2V-radius counts ≃4 spins (**b**). (**c**) In-plane components of spins; arrows are colored with the azimuthal angle $\theta \in [0^{\circ },360^{\circ })$, defined by the $s_{x}$ and $s_{y}$ components of the spin vector s and highlighting the in-plane spins rotational direction. A schematic representation of the two vortices and anti-bi-vortex together with a color wheel in (**g**) further helps the visualization of the planar orientation of spins. (**d**) Lateral view of the local eigenvectors for each M−X−M−X spin-ligand plaquette on the triangular *M*-net to help visualization of the non-coplanarity and non-collinearity in the exchange-tensor principal axes. (**e**) In-plane components of the red eigenvector pointing along the X−X direction for each six nearest-neighbor magnetic M−M pair. Specific case here concerns the two-ion anisotropy ($J^{S}$) estimated in monolayer NiI${}_{2}$ [40] (Table 1). Spins on the nearest-neighbour *M* sites of the central site orient according to the in-plane projection of the noncoplanar principal axes, as from the zoom on the spin texture of the A2V-core obtained by the MC simulations (**f**).

**Figure 8 nanomaterials-11-01873-f008:**
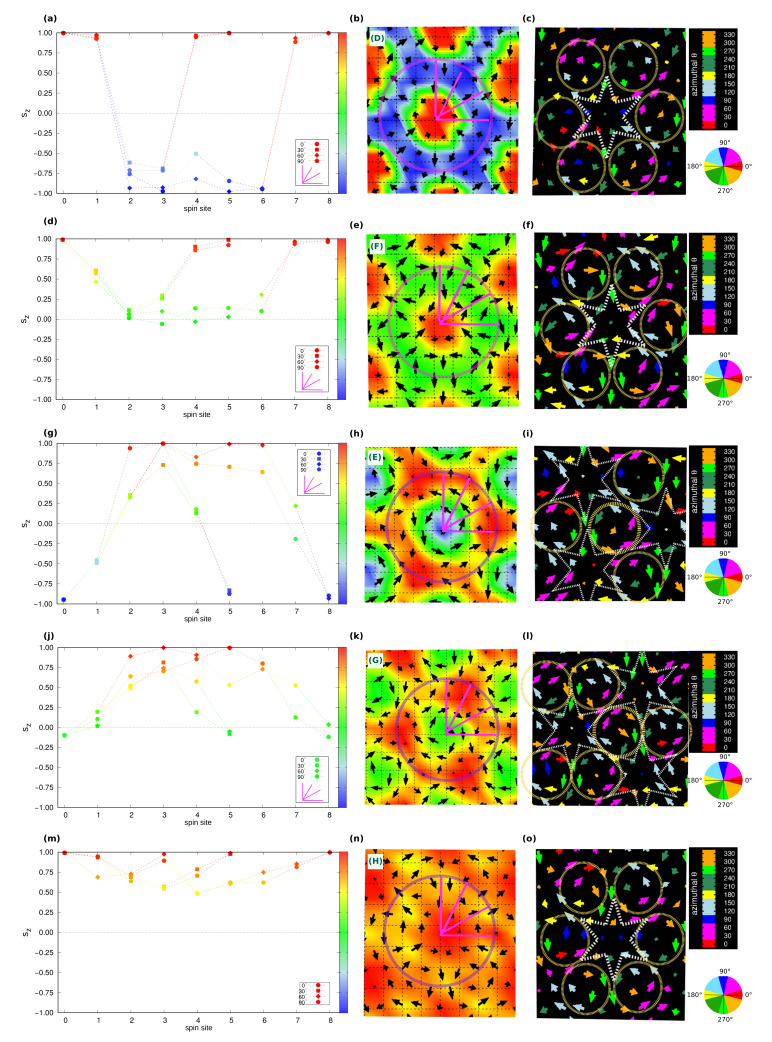
Profile of the out-of-plane component of spins ($s_{z}$) and planar spin texture with associated azimuthal angle defined by the $\left\{s_{x},s_{y}\right\}$ components, as in Figure 7a–c, for the A2SK-type lattice of the D-phase (**a**–**c**), the A2M-lattice of the F-phase (**d**–**f**), the Bloch-type SK-lattice of the E-phase (**g**–**i**), the meron lattice of the G-phase (**j**–**l**), and the trivial triple-**q** state of the H-phase (**m**–**o**).

**Table 1 nanomaterials-11-01873-t001:** Magnetic exchange coupling parameters and single-ion anisotropy, in terms of energy units (meV), for the NiI${}_{2}$ monolayer. In the adopted convention, negative (positive) values of the exchange parameters refer to FM (AFM) magnetic interaction, while the positive (negative) value of SIA ($A_{zz}$) indicates easy-plane (easy-axis) anisotropy.

$J^{1\mathrm{iso}}$	$J_{\mathrm{xx}}^{S}$	$J_{\mathrm{yy}}^{S}$	$J_{\mathrm{zz}}^{S}$	$J_{\mathrm{yz}}^{S}$	$J_{\mathrm{xz}}^{S}$	$J_{\mathrm{xy}}^{S}$	$J^{3\mathrm{iso}}$	$A_{\mathrm{zz}}$
−7.0	−1.0	+1.4	−0.3	−1.4	0.0	0.0	+5.8	+0.6

## Data Availability

Main results are reported in this article. All other data that support the findings discussed in this study are available from the corresponding author upon reasonable request.

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
