# Peer review of "Interplay between Single-Ion and Two-Ion Anisotropies in Frustrated 2D Semiconductors and Tuning of Magnetic Structures Topology"

_nanomaterials, 2021, doi:10.3390/nano11081873_

Round 1

Reviewer 1 Report

In the present manuscript, the authors investigate the effects of competing magnetic interactions in stabilizing different spin configurations. The most important results of the manuscript are as follows: 1) the authors constructed the phase diagram of a frustrated localized magnet on a two-dimensional centrosymmetric triangular lattice and predicted a rich variety of complex magnetic textures; 2) the authors underlined the interplay between the two-ion anisotropy (TIA) and the single-ion anisotropy (SIA) alongside with the effects of an external magnetic field 3) the authors found a dominant role played by the two-ion over the single-ion anisotropy in determining the planar spin texture and the single-ion over two-ion anisotropy in tuning the perpendicular spin components. The manuscript is well written and, in most cases, very clear. The topic of this study is certainly of interest for a large community interested in chiral magnetization states, and the new results add
important information to the field of research. I therefore think that the manuscript is suitable for publication.

At the same time, there are some ambiguous points that should be improved.

1) Sometimes I was confused with citations made in the present manuscript:
- The authors seem to cite only the newest papers on the subject overlooking actually the first papers that introduced some particular phenomena. For example, I think Ref. [20] must be cited when one introduces skyrmions with multiple topological charges, since skyrmions with Q=2 were introduced there. One may also recall Ref. [31], where a square lattice of merons was considered. Or Ref. [PRL 119, 207201 (2017)] where bound states of skyrmions and merons were considered. I also believe Ref. [Phys. Rev. B 96, 014423 (2017)] was one of the first papers to introduce bimerons, although there they were called nonaxisymmetric skyrmions.
- Some citations seem to have no connection to a particular statement. For
example, how is Ref. [17] related to a skyrmion lattice? (line 43)
- An interesting system to host skyrmion-like textures stabilized by interaction disorder are reentrant spin glasses as considered in Ref. [PRB 98, 014420 (2018).]
2) In the introduction, I would advise to mention that there is a long history behind stability of skyrmions. Skyrmion lattice was first experimentally observed within the A-phase of MnSi. And eventually, it was deduced that one may stabilize such a lattice, e.g., by thermal fluctuations or small anisotropies. In this sense, it sounds a bit ambiguous that “Conventional skyrmion lattices are usually observed in chiral magnets as a result of the interplay between Heisenberg and Dzyaloshinskii-Moriya”.
3) I think one needs closer comparison with already published results what would justify the novelty of the present manuscript. Indeed, the same phases have been already discovered in Ref. [39] and Ref. [35]. Then, I believe, one should point out what are the new effects found in the present manuscript?
4) In chiral magnetism, one usually considers a hierarchy of energy scales. There are basic interactions – exchange interaction, Dzyaloshinskii-Moriya interaction, and Zeeman energy – that are considered to stabilize all modulated phases. Then, different anisotropic contributions are considered as minor energy terms, which at the same time play an important role by shuffling different phases and defining their stability regions. I wonder whether the author would consider the same logic in the present manuscript? Then, I believe, even for zero anisotropies they should have all
the phases, but some of them are just metastable solutions.
5) It is known that the period of modulated phases changes upon action of an applied magnetic field or anisotropies. For example, it is a classical Dzyaloshinskii solution that the spiral state undergoes expansion in the field, i.e., the period of a spiral state eventually goes to infinity and the spiral transforms into a homogeneous state. Then, the question arises how do the authors take into account that the periods of all states depend on the field and values of the anisotropies. From all the figures in the manuscript, it seems that the size of the grid is always the same.
6) The authors might refer to Ref. [Phys. Rev. B 82, 052403 (2010)] on the influence of the uniaxial anisotropy on a skyrmions lattice and spiral states. The constructed phase diagram even includes some lability lines of different phases.
7) In this sense, I wonder, whether the phase H is simply an artefact of computational routines that is stabilized only in confined numerical grids?
8) Could the authors probably improve the discussion on topological charge
arithmetic? It is hard to understand that some skyrmion lattices have the topological charge 6, but in the Ref. [35], the same structure has the topological charge 2. Also, it is hard to understand how exactly such an arithmetic is done. For example, in line 231 it is claimed that Q = QA2V + 2QV = 2, but in the previous lines it is written that QA2V = −2 · pA2V and QV = +1 · pV, with 0 < pA2V < +1 and −1 < pV < 0. I do not see how to get Q=2 under such circumstances.
9) The authors could supplement the figures by additional sketches of cells over which they calculate their topological charge. Also, it is not clear why, e.g., in Fig. 3b, 3e, 4b etc., the authors use the color scale for topological charge densities that does not allow to discern any detail on the figures?
10) It seems that some modulated structures contain internal magnetic charges. In this sense, is the dipole-dipole interaction important? Can it be so easily omitted?

Reviewer 2 Report

Report for nanomaterials-1255594, “Tuning of magnetic structures topology via single-ion anisotropy and magnetic field in frustrated 2D semiconductors” by Danila Amoroso, Paolo Barone and Silvia Picozzi

 The authors present micro atomistic simulations investigating the effects of competing single-ion and two-ion anisotropies in a triangular lattice with strong magnetic frustration, as occurring in monolayers of van der Waals nickel dihalides. They analyzed the parameter space spanned by SIA and applied Bz field for a spin-lattice model whose parameters have been estimated for the monolayer of the prototypical semiconducting NiI2. They found a rich phase diagram comprising different magnetic phases, from topologically trivial single-(q) and triple-(q) states to topological triple-(q) states.

The present manuscript seems to be an extension to their work in Nat. Commun. 2020, 11, 5784 (Ref 35). The major results are already presented in this work. Nevertheless, are the result presented in their current manuscript - mainly an extension to the full parameter space - of interest and value to the general reader and therefore I can fully recommend publication of this manuscript in nanomaterials in its current form. There are only a few minor changes which the authors might want to consider:

  1. Please add details about the simulation as size and thickness of the cell in which the simulations have been performed. A pure reference to the Nature Communication. Paper is considering the total length of the paper hindering the easy reading of the paper. In addition, it would make the paper also more self-contained.
  2. In Ref. 31 the correct name is spelled Rößler.
